# IKK/NF-κB Inactivation by Salidroside via Targeting TNF-α for the Treatment of LPS-Induced Colitis

**DOI:** 10.3390/cimb47110896

**Published:** 2025-10-28

**Authors:** Qi Ouyang, Hao Zhou, Zixuan Yu, Hong Jiang, Chenhao Ji, Yijia Sun, Fang Zhou, Shuanglin Xiang, Xiang Hu

**Affiliations:** 1The National and Local Joint Engineering Laboratory of Animal Peptide Drug Development, College of Life Sciences, Hunan Normal University, Changsha 410081, China; oyq@hunnu.edu.cn (Q.O.); zhouhao1995@csmu.edu.cn (H.Z.); 202330233002@hunnu.edu.cn (Z.Y.); 202520143051@hunnu.edu.cn (H.J.); 202330233030@hunnu.edu.cn (C.J.); 202530151171@hunnu.edu.cn (Y.S.); 2College of Clinical Laboratory, Changsha Medical University, Changsha 410219, China; 3Engineering Research Center for Antibodies from Experimental Animals of Hunan Province, College of Life Sciences, Hunan Normal University, Changsha 410081, China; zhoufang@promab.cn

**Keywords:** salidroside, network pharmacology, TNF-α, IKK/NF-κB

## Abstract

Background: Tumor necrosis factor-alpha (TNF-α) serves as a central mediator of inflammation and represents key therapeutic target in inflammatory bowel disease (IBD). This study investigates the protective effects of salidroside (Sal) against inflammation and explores its underlying molecular mechanisms. Methods: We employed network pharmacology to identify potential targets of Sal. The anti-inflammatory effects of Sal were evaluated in LPS-Induced cellular models using NCM460 colonic epithelial cells and RAW264.7 macrophages, as well as in a murine model of acute colonic inflammation. Direct target engagement was confirmed through cellular thermal shift assay (CETSA) and co-immunoprecipitation (Co-IP). The mechanism was further elucidated via site-directed mutagenesis and analysis of the IKK/NF-κB signaling pathway. Results: Network pharmacology predicted TNF-α as a key target. Sal significantly attenuated LPS-Induced inflammation in vitro and ameliorated colitis symptoms in vivo. Notably, CETSA and Co-IP assays confirmed direct interaction between Sal and TNF-α. Mutagenesis studies identified Arg179, Lys188, and Tyr191 as critical residues for this binding. Mechanistically, Sal inhibited TNF-α-mediated activation of the IKK/NF-κB pathway and the subsequent production of pro-inflammatory cytokines. Conclusion: Our findings demonstrate that Sal alleviates inflammation by directly binding to TNF-α and suppressing the downstream NF-κB signaling cascade, thereby positioning it as a promising therapeutic candidate for TNF-α-driven inflammatory diseases.

## 1. Introduction

The initiation of inflammatory signaling cascades is intricately linked to the development of inflammatory diseases, often triggered by the disruption of epithelial barriers due to cell death [1]. This loss of barrier integrity facilitates microbial translocation and the release of pathogen-associated molecular patterns (PAMPs), which subsequently activate innate immune responses [2]. TNF-α is a central mediator of intestinal inflammation and a key cytokine in the pathogenesis of IBD, with elevated levels observed in affected patients. It exerts its effects through two specific receptors, TNFR1 and TNFR2, whose engagement activates various signaling pathways that regulate cell survival, death, and differentiation [3,4]. To establish a well-defined pathophysiological context for investigating the anti-TNF-α mechanism of salidroside (Sal), we employed the LPS-Induced murine colitis model. This model provides a direct and reproducible system characterized by a rapid onset of TNF-α-driven acute inflammation [5,6]. Pathologically, increased TNF-α expression compromises the mucosal barrier, thereby exacerbating inflammation and worsening disease prognosis [7,8]. The engagement of TNF-α with its receptors activates downstream pathways, including NF-κB and MAPK, leading to the production of key inflammatory mediators such as IL-1β, IL-6, and COX-2 [9,10]. Furthermore, the disruption of mucosal immunity—particularly the imbalance between pro-inflammatory Th17 cells and anti-inflammatory regulatory T cells (Tregs), coupled with alterations in the gut microbiota—perpetuates chronic inflammation in ulcerative colitis (UC) [11,12].

Consequently, TNF-α has emerged as a major therapeutic target. Monoclonal antibodies such as infliximab and adalimumab have demonstrated clinical efficacy [13]. However, the therapeutic potential of TNF antagonists is constrained by inadequate efficacy, with less than half of patients exhibiting robust responses, as well as [14,15]. Therefore, there is a critical need for safer and more effective agents that target the TNF-α pathway. Several natural compounds, such as glucoraphanin found in broccoli, have shown promising TNF-α inhibitory activity in recent studies [16,17]. Notably, their clinical translation is hindered by challenges related to oral bioavailability and target specificity [18].

In recent years, natural compounds derived from traditional Chinese medicine (TCM) have garnered increasing attention as potential anti-inflammatory agents [19]. Sal, a key bioactive compound from *Rhodiola rosea*, has demonstrated significant anti-inflammatory and immunomodulatory activities [20]. Critically, emerging evidence suggests that its activity may involve the modulation of TNF-α signaling [21,22]. Nevertheless, the direct molecular target of Sal in the pathogenesis of UC remains unknown. In this study, we demonstrate that Sal directly binds to TNF-α, and this interaction is critical for its inhibition of IKK/NF-κB signaling and its overall efficacy in mitigating experimental colitis.

## 2. Materials and Methods

### 2.1. Chemical and Reagents

Salidroside (Macklin, Shanghai, China), LPS (Sigma, St. Louis, MO, USA), and Dexamethasone (AbMole, Houston, TX, USA) were commercially obtained. The Mouse ELISA Kit was from Zikerbio (Shenzhen, China). Primary antibodies against NF-κB p65 (HUABio, Hangzhou, China, 1:1000 dilution), p-NF-κB p65 (HUABio, Hangzhou, China, 1:1000 dilution), IKKα/β (Selleck, Houston, TX, USA, 1:1000 dilution), p-IκBα (Ser36) (Selleck, Houston, TX, USA, 1:10,000 dilution), IL-1β (Immunoway, Plano, TX, USA, 1:1000 dilution), TNF-α (Immunoway, Plano, TX, USA, 1:1000 dilution), F4/80 (ZenBio, Chengdu, China, 1:100 dilution), CD3 (ZenBio, Chengdu, China, 1:100 dilution) and CD4 (ZenBio, Chengdu, China, 1:100 dilution) were used. HRP-conjugated secondary antibodies were from Abbkine (Wuhan, China). Reagents for qPCR (HiScript® II Q RT SuperMix, SYBR Green Master Mix; Younggen Bio, Nanjing, China) and RNA extraction (RNAiso Easy; Takara, Beijing, China) were applied as directed..

### 2.2. Cell Culture and Treatment

The human normal colonic epithelial cell line NCM460 and the murine macrophage cell line RAW264.7 were selected to model the colonic epithelial barrier and innate immune responses, respectively, which are two critical aspects of ulcerative colitis pathophysiology. NCM460 and RAW264.7 cells were maintained in DMEM or RPMI 1640 medium, respectively, each supplemented with 10% fetal bovine serum, at 37 °C in a 5% CO_2_ atmosphere. To assess cellular viability, cells were seeded at a density of 1 × 10^5^ cells/well in a 96-well plate and treated with various concentrations of Sal, LPS, or Dex (dexamethasone) for 24 h, followed by an MTT assay. For mechanistic investigations, cells were seeded at a density of 8 × 10^5^ cells/well in a 6-well plate. The normal control (NC) group was left untreated. Cells were pretreated with Sal (200 μM) for 24 h prior to exposure to LPS (2 μg/mL) for an additional 24 h. In parallel experiments designed to evaluate the effects of individual agents, cells were treated with PBS (control), Sal, or Dex alone for 24 h before harvest for RT-qPCR and Western blot analysis. To specifically examine TNF-α-induced signaling, cells were seeded at a density of 1 × 10^5^ cells/well in a 6-well plate and stimulated with recombinant TNF-α (1 μg/mL) for 24 h prior to Western blot analysis.

### 2.3. Animals and Treatment

Male C57BL/6J mice (6–8 weeks old) were purchased from Silaike Jingda (Beijing, China). All experimental procedures were approved by the Animal Ethics Committee of Hunan Normal University (2024-775). The mice were maintained under standard housing conditions with a 12 h light/dark cycle, controlled temperature and humidity, and were provided ad libitum access to standard laboratory diet and water.

The LPS-Induced murine colitis model was employed to specifically elicit a robust, TNF-α-dominated acute inflammatory response, making it suitable for directly testing the anti-TNF-α mechanism of Sal. A total of 12 mice were randomly divided into four experimental groups (*n* = 3): (1) the Control group, which received an equivalent volume of vehicle (PBS); (2) the LPS group, which received a single intraperitoneal injection of 1 mg/kg LPS; (3) the Sal group, which was administered 4 mg/kg/day of Sal via intraperitoneal injection daily; and (4) the Sal + LPS group, which received daily Sal (4 mg/kg/day) following a single LPS (1 mg/kg) injection.

Body weight and general health status were monitored daily. All mice were euthanized 48 h after the LPS injection. Colon tissues were promptly collected, measured, and processed for subsequent histological and molecular analyses.

### 2.4. Western Blot

Proteins were extracted using RIPA buffer (CWBIO, Taizhou, China). 10 μg of protein per sample were separated by SDS-PAGE, transferred to PVDF membranes, and incubated with primary antibodies overnight at 4 °C. After incubation with HRP-conjugated secondary antibodies, bands were visualized using an ECL system (Suzhou, China). Quantitative densitometric analysis was performed on the immunoreactive bands using ImageJ software (version 1.52a) (National Institutes of Health, USA). To account for variations in protein loading, the optical density of each target protein band was normalized to that of the internal control (β-actin) within the same sample. Statistical analysis was conducted on data from at least three independent replicates.

### 2.5. Histopathology and Immunohistochemistry (IHC)

Colon sections were stained with H&E and PAS for pathological assessment. The severity of colitis and the number of goblet cells were evaluated by two independent researchers blinded to group allocation, using a well-established scoring system that considers the extent of inflammatory cell infiltration, tissue damage, and crypt architecture loss. For IHC, sections were incubated with anti-F4/80, anti-CD3, or anti-CD4 antibodies, followed by HRP-conjugated secondary antibodies and DAB development. The number of positive macrophages and T cells, expressed as a percentage of the stained area, was quantified in at least 5 randomly selected fields per sample using ImageJ software or an automated image analysis system.

### 2.6. Quantitative Real Time-PCR (qPCR)

Total RNA was isolated from colon tissues or cultured cells using TRIzol reagent (Takara, Beijing, China) according to the manufacturer’s protocol. Equal amounts of RNA were reverse transcribed into cDNA using HiScript^®^ II Q RT SuperMix. Quantitative real-time PCR was performed on a Bio-Rad CFX Real-Time PCR System with SYBR Green qPCR Master Mix. Gene expression levels were normalized to β-actin and analyzed using the 2^−ΔΔCT method. All primer sequences used in this study are listed in Appendix A.

### 2.7. Network Pharmacology and Molecular Docking

The Sal structure was obtained from PubChem for all analyses. SwissTargetPrediction was used to predict its protein targets in Homo sapiens (probability > 0) (Appendix A). UC-related genes were collected from GeneCards (relevance score: medium). A Venn diagram was employed to identify shared targets, which were then used to construct a protein–protein interaction (PPI) network via the STRING database (minimum interaction score: 0.4). Core targets from the network underwent functional enrichment analysis (GO and KEGG) with a significance threshold of adjusted *p*-value < 0.05. The heatmap was generated using https://www.bioinformatics.com.cn (last accessed on 10 December 2024), an online platform for data analysis and visualization [23].

For molecular docking, the 3D structure of Sal was obtained from PubChem. The protein structures of TNF-α and TNFAIP1 were modeled using AlphaFold 3.0, while the crystal structures of COX-2 (PDB ID: 1CX2) and ZO-1 (PDB ID: 3LH5) were retrieved from the RCSB Protein Data Bank. Docking simulations were performed with AutoDock Vina 1.1.2 to evaluate the binding affinities (kcal/mol) and interaction modes between Sal and these target proteins (Appendix A).

### 2.8. Plasmid Construction, Mutagenesis, and Transfection

Wild-type and mutant (ARG179, LYS188, TYR191) TNF-α plasmids were constructed and verified by sequencing. NCM460 cells were transfected using Lipofectamine 3000 and harvested after 36 h for analysis.

### 2.9. Cellular Thermal Shift Assay (CETSA)

NCM460 cell lysates, containing approximately 1 × 10^7^ cells, were incubated with 200 μM Sal for 30 min at 4 °C. Variable temperature experiments were conducted at 37, 45, 52, and 62 °C for 5 min. The supernatants obtained by centrifugation were analyzed by Western blot using a TNF-α antibody. The vehicle was used as the control group.

### 2.10. Immunoprecipitation (IP)

Lysates prepared from NCM460 cells from approximately 1 × 10^7^ cells were treated with 200 μM at 37 °C for 24 h, then incubated overnight at 4 °C with TNF-α or IgG antibody and protein A/G magnetic beads. The resulting protein was analyzed by Western blot.

### 2.11. Statistical Analysis

Data are presented as the mean ± standard deviation (S.D.). Statistical analyses were performed using GraphPad Prism 9.0 software. For experiments with one independent variable (e.g., different drug treatments under the same condition), data were analyzed by one-way ANOVA followed by Tukey’s post hoc test for multiple comparisons. For experiments with two independent variables (e.g., the effects of both LPS and Sal treatment), data were analyzed by two-way ANOVA followed by Tukey’s post hoc test. A *p*-value of less than 0.05 (*p* < 0.05) was considered statistically significant. The sample size for each experiment (*n* = 3 for in vivo studies, *n* = 3 for in vitro replicates) is provided in the corresponding figure legends. The animals group size (*n* = 3) was chosen based on established protocols for preliminary mechanistic studies in this model, which consistently yield significant and interpretable results with this sample size.

## 3. Results

### 3.1. Salidroside Alleviates Inflammation on LPS-Induced Mice

To evaluate the therapeutic effect of Sal on acute colonic inflammation, a mouse model was established via a single intraperitoneal injection of LPS. The Sal-treated groups were compared with the model group to assess efficacy. The dose of Sal used in this study was determined based on the existing literature, which demonstrated both efficacy and an absence of adverse effects in relevant models [24,25]. The animal modeling and treatment protocol is illustrated in Figure 1A. Administration of Sal significantly ameliorated LPS-Induced body weight loss (Figure 1B) and colonic shortening (Figure 1C,D). The maintenance of normal gut function is regulated by a tightly controlled balance of mucus secretion [26]. Histopathological analysis corroborated these findings. H&E staining revealed that LPS challenge induced severe colonic damage, characterized by crypt disintegration and inflammatory cell infiltration, which was markedly attenuated by Sal treatment (Figure 1E). Furthermore, PAS staining demonstrated that Sal effectively restored the LPS-Induced depletion of mucin-producing goblet cells (Figure 1F), suggesting a role in preserving the mucosal barrier. Having established the protective effects of Sal on colonic pathology, we next investigated the underlying immune responses. Consistent with the observed histopathological improvement, Sal treatment led to a significant reduction in the colonic mRNA levels of key pro-inflammatory cytokines, including TNF-α, IL-1β, and IL-6 (Figure 1G). To directly assess immune cell infiltration, we performed immunohistochemistry. The results showed a substantial influx of F4/80^+^ macrophages, CD3^+^ T cells, and CD4^+^ helper T cells into the colon upon LPS challenge, which was notably suppressed by Sal administration, as demonstrated by immunohistochemical staining (Figure 1H).

Taken together, these findings indicate that Sal alleviates LPS-Induced colitis through a dual mechanism: restoring mucosal integrity and reducing the recruitment of pro-inflammatory immune cells, thereby curtailing the production of pro-inflammatory cytokines.

### 3.2. Effects of Salidroside on NCM460 and RAW264.7 Cells Induced by LPS

To elucidate the direct anti-inflammatory effects of Sal, we established in vitro models using LPS-Induced NCM460 colonic epithelial cells and RAW264.7 macrophages. Initially, we determined the non-cytotoxic concentration ranges for Sal, LPS, and Dex through cytotoxicity assays in NCM460 cells. Based on prior literature and our initial cytotoxicity screening [25], subsequent experiments in NCM460 cells were conducted using 200 μM Sal, 2 μg/mL LPS, and 2 μg/mL Dex (Figure 2A). In the NCM460 epithelial model, ELISA assay results indicated that Sal treatment significantly reduced the LPS-Induced secretion of TNF-α, IL-1β, and IL-6 into the cell supernatant (Figure 2B). This effect was corroborated at the transcriptional level, as Sal also decreased the mRNA expression of these cytokines (Figure 2C). We next validated this anti-inflammatory activity in RAW264.7 macrophage cells. A parallel cytotoxicity assessment guided the selection of concentrations of 300 μM Sal, 2 μg/mL LPS, and 2 μg/mL Dex for this cell line (Figure 2D). Consistent with the findings in epithelial cells, Sal effectively suppressed both the release (Figure 2E) and the mRNA expression levels (Figure 2F) of TNF-α, IL-1β, and IL-6 in LPS-stimulated RAW264.7 cells.

In summary, these data demonstrate that Sal directly inhibits the expression and secretion of key pro-inflammatory cytokines in both intestinal epithelial cells and immune cells.

### 3.3. Salidroside Ameliorates Cell Inflammation by Inhibiting the IKK/NF-κB Signaling Pathway

To elucidate the potential mechanism underlying the anti-inflammatory effects of Sal, we investigated the activation status of the IKK/NF-κB signaling pathway, a key regulator of pro-inflammatory cytokine production, in both NCM460 and RAW264.7 cells following LPS induction. The protein expression and phosphorylation levels of critical signaling molecules were analyzed using Western blotting. In NCM460 cells, LPS challenge markedly increased the phosphorylation levels of IKKα/β, IκBα, and NF-κB p65, indicating robust activation. Treatment with Sal significantly attenuated these LPS-Induced phosphorylation events (Figure 3A,B). Similarly, in RAW264.7 cells, Sal effectively suppressed the LPS-triggered phosphorylation of IKKα/β, IκBα, and NF-κB p65 (Figure 3C,D). The total protein levels of these components remained largely unchanged across all groups. Notably the inhibitory effect of Sal was comparable to that of the positive control, the Dex group.

These results collectively demonstrate that Sal effectively inhibits the activation of the IKK/NF-κB pathway in both intestinal epithelial cells and macrophages, thereby providing a mechanistic explanation for its suppression of pro-inflammatory cytokine production.

### 3.4. Salidroside Ameliorates Colitis by Inhibiting the TNF-α Signaling Pathway

Sal was gathered from the SwissTargetPrediction and TCMSP databases, leading to the identification of 101 predictive proteins (Appendix A) [27]. Additionally, we collected 66 IBD-associated proteins from the OMIM database, 2662 IBD-associated protein genes from the GeneCards, and 29 IBD-associated proteins from the DISEASE database. After removing duplicates, 689 disease-related protein genes remained. A Venn diagram was created using these 689 disease proteins and the 101 active compound proteins (Figure 4A), resulting in the identification of 13 overlapping proteins, which were selected as the primary genes for further study. To fully understand the potential mechanism by which Sal treats LPS-Induced colon inflammation, the gene names of the 13 Sal anti-UC proteins were imported into the STRING database and Cytoscape 3.9.1, with the minimum required interaction score set at 0.8. Utilizing the STRING network and based on protein degree, the PPI network was constructed (Figure 4B). GO enrichment analysis included 67 biological process (BP) terms, 4 cellular component (CC) terms, and 4 molecular function (MF) terms. The top 12 significantly enriched BP terms associated with the pathogenesis of inflammation are presented, along with the complete lists of CC and MF terms (Figure 4C). KEGG enrichment analysis identified 75 pathways (*p* < 0.05), with 12 pathways highlighted (Figure 4D). According to the KEGG analysis, the effects of Sal were primarily related to immunological and disease-associated pathways, with disease-related pathways mainly involving cancer, infection, and inflammatory bowel disease. Immune-related pathways primarily included the IL-17 and TNF signaling pathways.

### 3.5. Arg179, Lys188 and Tyr191 Played Roles in the Binding of Sal with TNF-α

Molecular docking was employed to investigate the potential binding modes and affinities of Sal toward three candidate targets: COX-2 (Appendix A), ZO-1 (Appendix A) and TNF-α. The docking results revealed that Sal could form stable complexes with all three proteins (Figure 5A and Appendix A), with the calculated binding free energies presented in Appendix A. Despite the docking scores, our previous in vivo and in vitro data consistently demonstrated that Sal potently suppressed the expression and secretion of TNF-α (Figure 1G and Figure 2B,E). This pronounced phenotypic effect led us to hypothesize that TNF-α might be a direct target, prompting us to investigate the functional interaction between Sal and TNF-α through a series of mechanistic experiments. Co-IP assays revealed that Sal treatment reduced TNF-α protein levels in immunoprecipitated complexes, suggesting a direct interaction, similar results were obtained from CETSA (Figure 5B and Appendix A). Meanwhile, we used TNF-α to induce inflammation in NCM 460 and RAW264.7 cells. Sal treatment significantly suppressed the expression of TNF-α itself, as well as downstream inflammatory mediators such as COX-2 (Figure 5C–F). Subsequently, we mutated Arg179, Lys188, and Tyr191 to Arg179Ala, Lys188Ala, and Tyr191Ala to explore their roles. Western blot analysis revealed that the mutations resulted in reduced TNF-α expression even in the absence of Sal (Figure 5G and Appendix A). Further CETSA experiments demonstrated that while Sal decreased the stability of wild-type TNF-α, all mutants exhibited inherently lower stability regardless of Sal treatment (Figure 5H and Appendix A). Although Sal affected TNF-α expression, the CETSA results collectively indicate that residues Arg179, Lys188, and Tyr191 are critical for the direct binding and functional antagonism of TNF-α by Sal.

Collectively, these findings indicate that Sal directly binds to TNF-α, promoting its destabilization and thereby attenuating TNF-α-mediated inflammatory signaling. Furthermore, the results identify Arg179, Lys188, and Tyr191 as essential residues for the functional interaction between Sal and TNF-α.

## 4. Discussion

This study demonstrates that Sal mitigates LPS-Induced acute colonic inflammation by directly targeting TNF-α and suppressing the downstream IKK/NF-κB signaling pathway. Our data consistently show that Sal treatment restores intestinal barrier integrity, reduces the infiltration of inflammatory immune cells (F4/80, CD3, CD4), and decreases the production of key pro-inflammatory cytokines (TNF-α, IL-1β, IL-6) in both in vivo and in vitro models. Notably, the reduction in CD4+ T cell and macrophage infiltration is particularly significant within the context of mucosal immunity, as the balance between regulatory T cells (Treg) and T helper 17 (Th17) cells in the Treg/Th17 axis is a central driver of acute colonic inflammation [28]. Crucially, molecular docking and mutagenesis studies provide mechanistic insight by identifying Arg179, Lys188, and Tyr191 as critical residues mediating the direct interaction between Sal and TNF-α.

The activation of the IKK/NF-κB pathway represents a canonical response to pro-inflammatory stimuli, including LPS and TNF-α, which ultimately drives the transcription of genes central to inflammation [29]. While previous studies have documented the association between Sal and reduced NF-κB activation, our data provide a more definitive mechanistic link [30]. We demonstrate that Sal acts upstream by directly interacting with TNF-α, an interaction localized to specifically at residues that likely impedes TNF-α’s capacity to initiate the inflammatory signaling cascade.

The search for novel TNF-α inhibitors, particularly orally available small molecules, remains a critical focus in the treatment of inflammatory diseases [31]. Although biological agents have revolutionized care, limitations such as immunogenicity, high costs, and variable patient responses persist. This has spurred the exploration of alternative strategies, including drug repurposing and the development of novel targeted delivery systems. For instance, recent studies have demonstrated the efficacy of repurposing antiviral drugs like remdesivir to modulate colonic inflammation, as well as the success of advanced lipid carriers for the targeted delivery of anticoagulants in UC, highlighting promising non-biologic avenues for treatment [32,33]. Indeed, network meta-analyses confirm the high efficacy of anti-TNF-α agents in challenging clinical settings scenarios, such as preventing recurrence in Crohn’s disease, which underscores the urgent need for alternatives that can circumvent overcome these drawbacks [34]. Natural products like curcumin and resveratrol have shown promise as TNF-α inhibitors, and advanced target identification techniques are accelerating their discovery [35,36]. Our work positions Sal within this promising category as a direct small-molecule antagonist of TNF-α, offering a distinct approach compared to monoclonal antibodies.

In summary, our findings demonstrate that Sal inhibits the activation of the IKK/NF-κB pathway, thereby alleviating inflammation induced by LPS. This effect includes the restoration of intestinal barrier integrity and a reduction in immune cell infiltration, specifically F4/80, CD3, and CD4 positive cells. Furthermore, we identified Sal as a potential therapeutic agent for TNF-α inhibition, offering a broader perspective for the treatment of IBD and suggesting Sal as a promising candidate for the development of novel TNF-α inhibitors.

## 5. Conclusions

In this study, we demonstrated the molecular mechanism by which Sal exerts its effects against LPS-Induced colonic inflammation through an integrated strategy. Our findings indicate that Sal effectively alleviates inflammatory responses and restores intestinal mucosal barrier integrity in vivo. Additionally, in vivo experiments confirmed that Sal can suppress inflammation via the IKK/NF-κB signaling pathway. Furthermore, we identified that Sal interacts with amino acids Arg179, Lys188, and Tyr191 to inhibit TNF-α activity, thereby reducing the production of inflammatory factors.

## Figures and Tables

**Figure 1 cimb-47-00896-f001:**
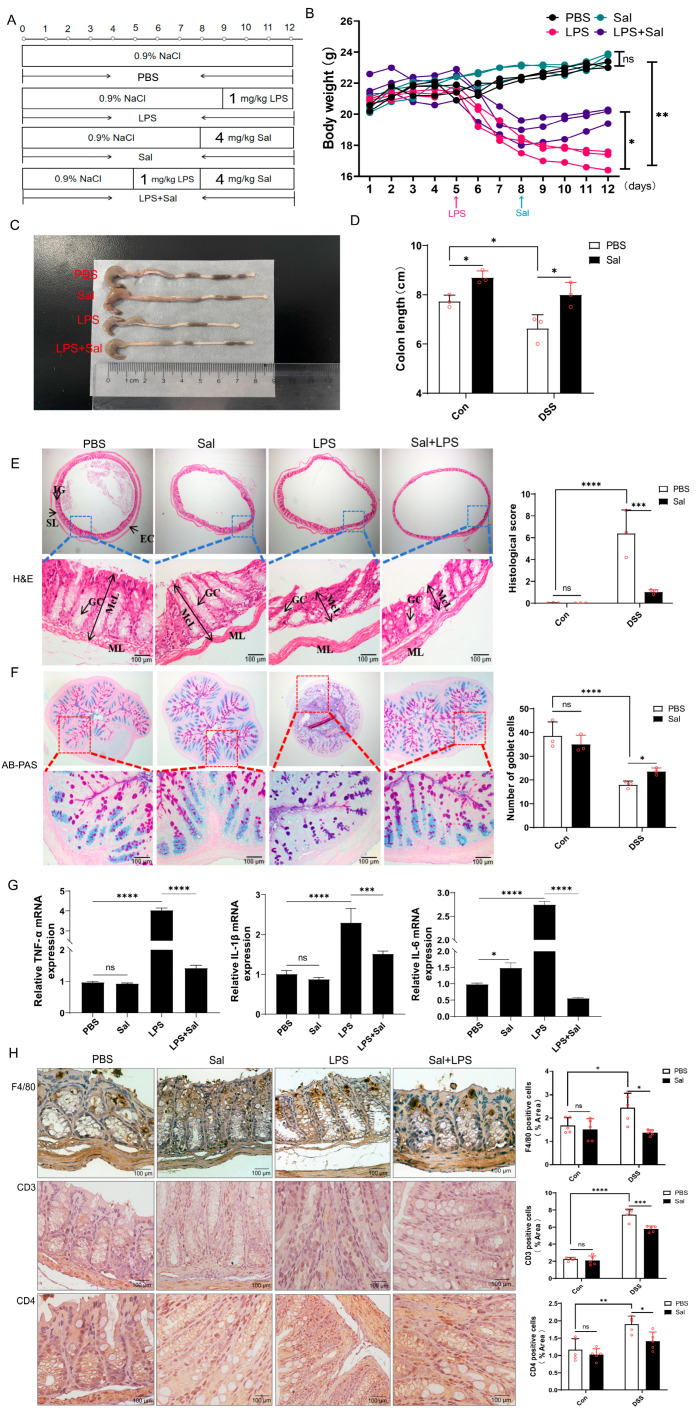
Sal alleviates inflammation on LPS-Induced mice. (**A**). Modeling and treatment cycle. (**B**). The body weight of various treatment groups. PBS vs. Sal, ns: no significance; PBS vs. LPS, ** *p* < 0.01; LPS vs. LPS + Sal, * *p* < 0.05. (**C**). Photographs of colons in various treatment groups. (**D**). The length of colons in various treatment groups. PBS vs. Sal, * *p* < 0.05; PBS vs. LPS, * *p* < 0.05; LPS vs. LPS + Sal, * *p* < 0.05. (**E**). Colon sections were stained with H&E and histological score. PBS vs. Sal, ns: no significance; PBS vs. LPS, **** *p* < 0.001; LPS vs. LPS + Sal, *** *p* < 0.005. (**F**). Colon sections were stained with AB-PAS and the number of goblet cells. PBS vs. Sal, ns: no significance; PBS vs. LPS, **** *p* < 0.001; LPS vs. LPS + Sal, * *p* < 0.05. (**G**). The mRNA levels of TNF-α, IL-1β and IL 6. PBS vs. Sal, ns: no significance; PBS vs. LPS, **** *p* < 0.001; LPS vs. LPS + Sal, *** *p* < 0.005 or **** *p* < 0.001. (**H**). Representative immunohistochemistry images of colon sections stained for the macrophage marker F4/80, the pan-T cell marker CD3, and the helper T cell marker CD4. Sal treatment reduced the infiltration of these immune cells induced by LPS. Scale bar, 100 μm. PBS vs. Sal, ns: no significance; PBS vs. LPS, * *p* < 0.05; ** *p* < 0.01; **** *p* < 0.001; LPS vs. LPS + Sal, * *p* < 0.05 or *** *p* < 0.005.

**Figure 2 cimb-47-00896-f002:**
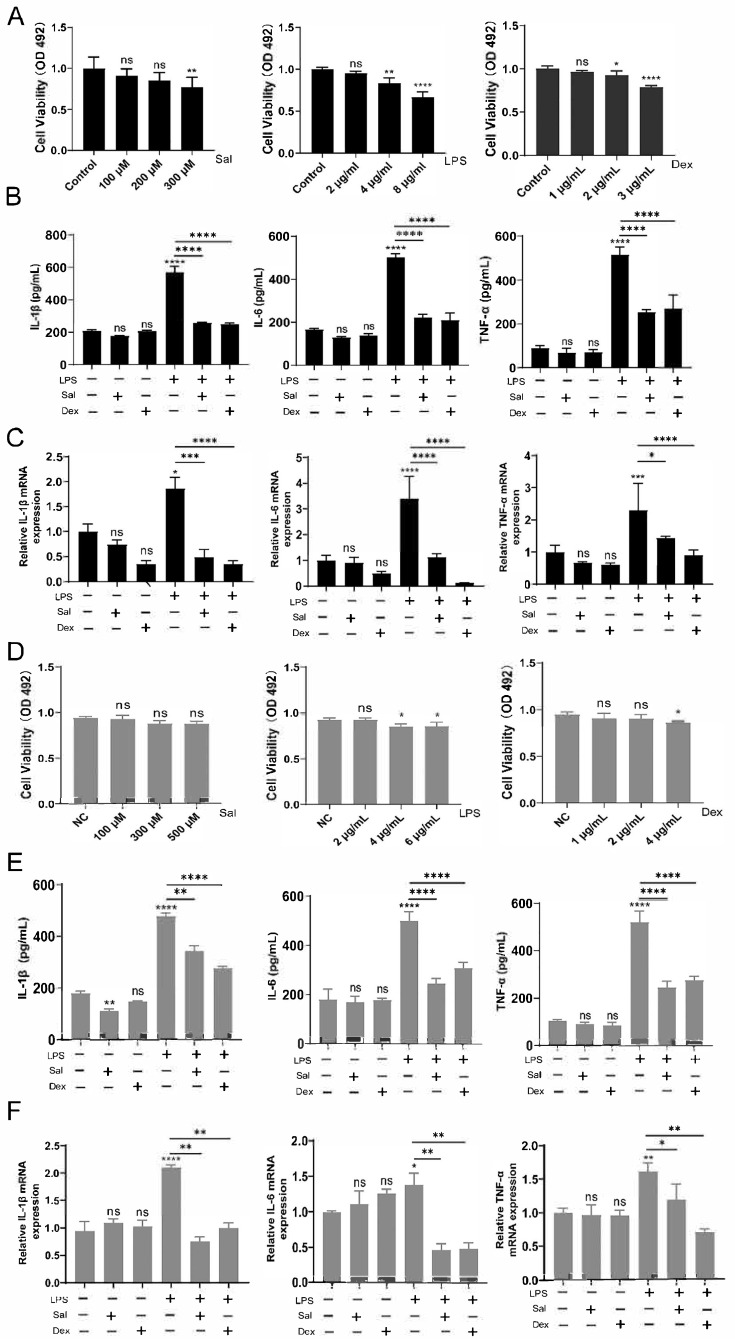
Effects of Sal on NCM460 and RAW264.7 cells induced by LPS. (**A**). Cell viability of different doses of Sal, LPS and Dex in NCM 460 cells. Different concentration of Sal, LPS or Dex vs. control, ns: no significance; * *p* < 0.05; ** *p* < 0.01; **** *p* < 0.001. (**B**). Measurement of proinflammatory (TNF-α, IL-1β and IL-6) factors produced from different treatment in NCM 460 cells by ELISA assay. PBS vs. 200 μM Sal or 2 μg/mL Dex, ns: no significance; PBS vs. 2 μg/mL LPS, **** *p* < 0.001; LPS vs. LPS + Sal, **** *p* < 0.001. LPS vs. LPS + Dex, **** *p* < 0.001. (**C**). The mRNA levels of TNF-α, IL-1β and IL-6 with different treated in NCM 460 cells. PBS vs. Sal or Dex, ns: no significance, PBS vs. LPS, * *p* < 0.05, *** *p* < 0.005, **** *p* < 0.001; LPS vs. LPS + Sal, * *p* < 0.05, *** *p* < 0.005, **** *p* < 0.001. LPS vs. LPS + Dex, **** *p* < 0.001. (**D**). Cell viability of different doses of Sal, LPS and Dex in RAW264.7 cells. Different concentration of Sal, LPS or Dex vs. control, ns: no significance; * *p* < 0.05. (**E**). Measurement of proinflammatory (TNF-α, IL-1β and IL-6) factors produced from different treatment in RAW264.7 cells by ELISA assay. PBS vs. Sal, ** *p* < 0.05; PBS vs. Dex, ns: no significance; PBS vs. LPS, **** *p* < 0.001; LPS vs. LPS + Sal, ** *p* < 0.01, **** *p* < 0.001; LPS vs. LPS + Dex, **** *p* < 0.001. (**F**). The mRNA levels of TNF-α, IL-1β and IL-6 with different treated in RAW264.7 cells. PBS vs. Sal or Dex, ns: no significance; PBS vs. LPS, * *p* < 0.05, ** *p* < 0.01, **** *p* < 0.001; LPS vs. LPS + Sal, * *p* < 0.05, ** *p* < 0.01; LPS vs. LPS + Dex, ** *p* < 0.05.

**Figure 3 cimb-47-00896-f003:**
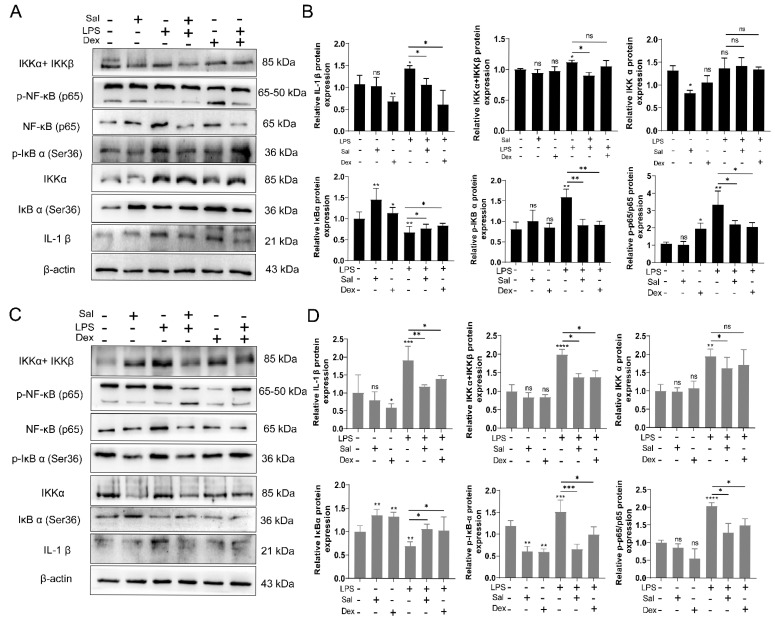
Salidroside ameliorates cell inflammation by inhibiting the IKK/NF-κB signaling pathway. (**A**). Representative blot images of IKKα + IKKβ, p-NF-κB, IκBα (Ser36), p-IκBα (Ser36) and IL-1β in NCM 460 cells. (**B**). Quantitative analysis of IKKα + IKKβ, p-NF-κB, IκBα (Ser36), p-IκBα (Ser36) and IL-1β in NCM 460 cells. NC vs. Sal, ns, no significance, * *p* < 0.05, ** *p* < 0.01; NC vs. Dex, ns, no significance, * *p* < 0.05, ** *p* < 0.01; NC vs. LPS, ns, no significance, * *p* < 0.05, ** *p* < 0.01; LPS vs. LPS +Sal, ns, no significance, * *p* < 0.05, ** *p* < 0.01; LPS vs. LPS +Dex, ns, no significance, * *p* < 0.05, ** *p* < 0.01. (**C**). Representative blot images of IKKα + IKKβ, p-NF-κB, IκBα (Ser36), p-IκBα (Ser36) and IL-1β in RAW264.7 cells. (**D**). Quantitative analysis of IKKα + IKKβ, p-NF-κB, IκBα (Ser36), p-IκBα (Ser36) and IL-1β in RAW264.7 cells. The values represent the mean ± S.D. NC vs. Sal, ns, no significance, ** *p* < 0.01; NC vs. Dex, ns, no significance, * *p* < 0.05, ** *p* < 0.01; NC vs. LPS, ** *p* < 0.01, *** *p* < 0.005, **** *p* < 0.001; LPS vs. LPS +Sal, * *p* < 0.05, ** *p* < 0.01, *** *p* < 0.005; LPS vs. LPS + Dex, ns, no significance, * *p* < 0.05.

**Figure 4 cimb-47-00896-f004:**
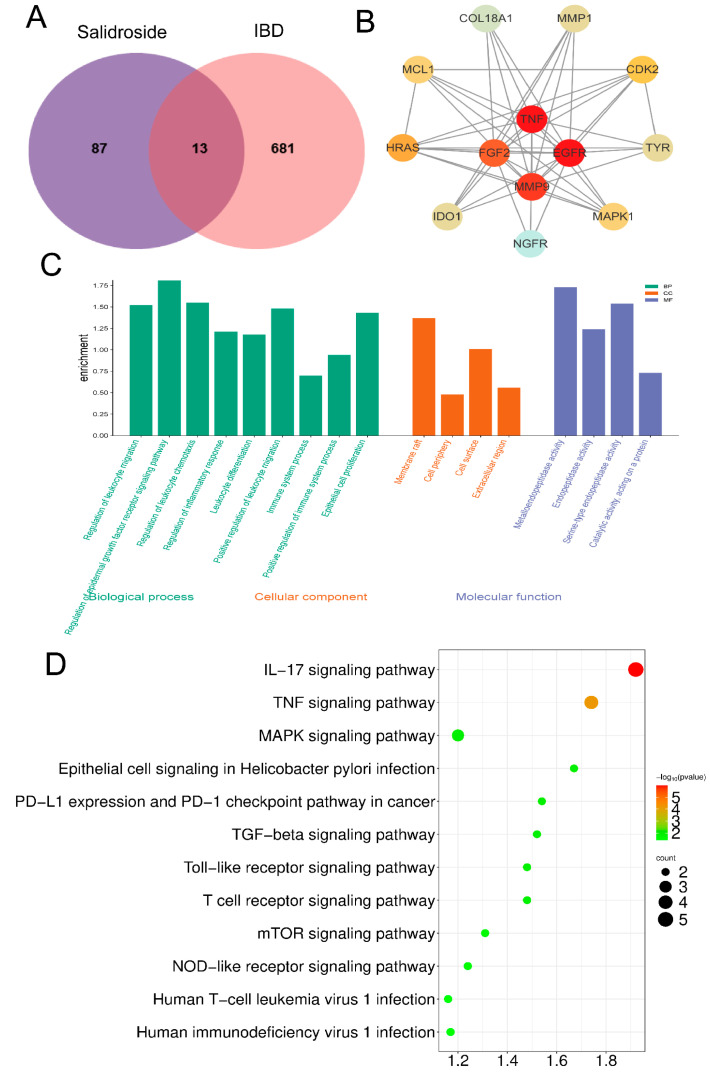
Sal ameliorates colitis by inhibiting the TNF-α signaling pathway. (**A**). A Venn diagram depicting the overlap between candidates and predicated targets of Sal in the context of IBD. (**B**). The PPI network of targeted proteins was generated via the STRING database and Cytoscape. (**C**). A bar graph illustrating the results of GO enrichment analysis, including biological process (BP), cell component (CC) terms, molecular function (MF). (**D**). The top 12 pathways related to the effect of Sal against UC.

**Figure 5 cimb-47-00896-f005:**
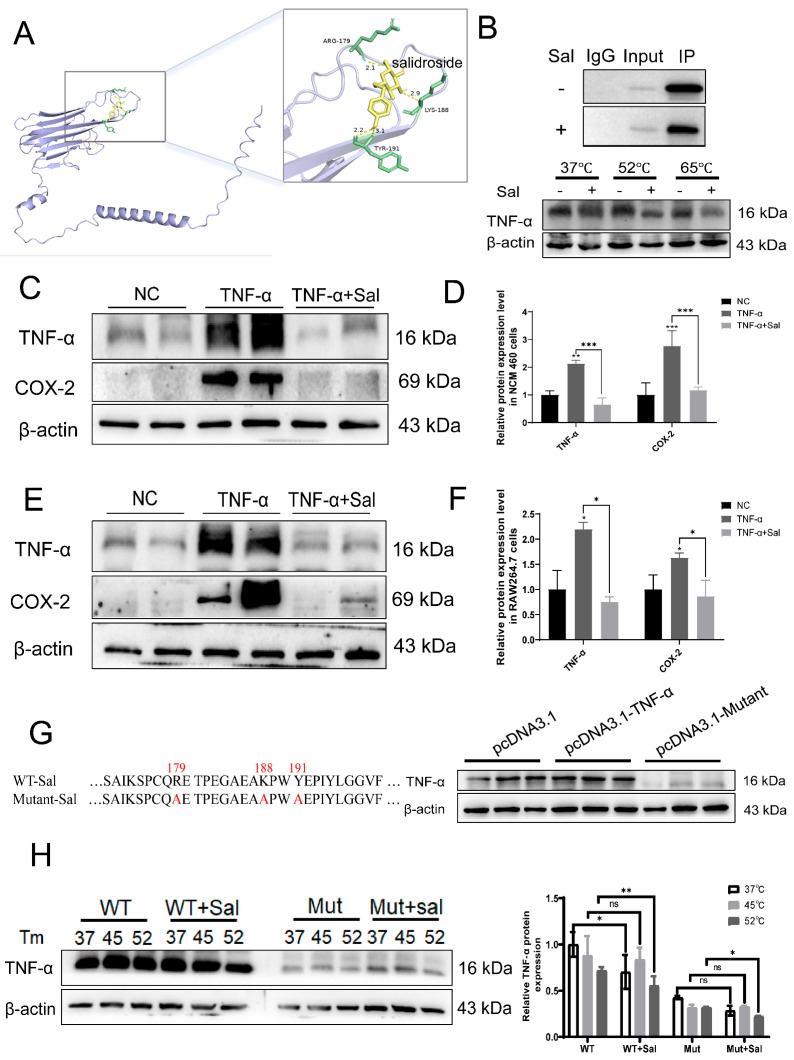
Arg179, Lys188 and Tyr191 played roles in the binding of Sal with TNF-α. (**A**). Molecular docking of Sal with TNF-α. (**B**). Co-immunoprecipitation (Co-IP) and CETSA of TNF-α protein stability in NCM 460 cells treated with Sal. (**C**). Sal abolished the anti-inflammatory (TNF-α, COX-2) in 1 μg/mL TNF-α-induced NCM 460 cells. (**D**). Quantitative analysis of TNF-α, COX-2 in NCM 460 cells. NC vs. TNF-α, ** *p* < 0.01, *** *p* < 0.005; TNF-α vs. TNF-α + Sal, *** *p* < 0.005. (**E**). Sal abolished the anti-inflammatory (TNF-α, COX-2) in 1 μg/mL TNF-α-induced RAW264.7 cells. (**F**). Quantitative analysis of TNF-α, COX-2 in RAW264.7 cells. NC vs. TNF-α, * *p* < 0.05; TNF-α vs. TNF-α + Sal, * *p* < 0.05. (**G**). Partial protein sequence alignment of wild-type (WT) and mutant (Arg179Ala, Lys188Ala, Tyr191Ala) TNF-α. Mutated residues are highlighted in red. Western blot analysis of TNF-α protein expression in NCM 460 cells transfected with the empty pcDNA3.1 vector, pcDNA3.1-TNF-α WT, or the respective pcDNA3.1-TNF-α mutants (Arg179Ala, Lys188Ala, Tyr191Ala). β-actin served as a loading control. (**H**). Arg179Ala, Lys188Ala, Tyr191Ala abolished the binding of salidroside with TNF-α. The values represent the mean ± S.D. WT vs. WT +Sal, ns, no significance, * *p* < 0.05, ** *p* < 0.001. Mutant vs. Mutant +Sal, ns, no significance, * *p* < 0.05.

## Data Availability

The original contributions presented in this study are included in the article/Appendix A. Further inquiries can be directed to the corresponding authors.

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
