# Peer review of "IKK/NF-κB Inactivation by Salidroside via Targeting TNF-α for the Treatment of LPS-Induced Colitis"

_cimb, 2025, doi:10.3390/cimb47110896_

Round 1

Reviewer 1 Report

Comments and Suggestions for Authors

1️⃣ Critical Review of Contribution & Novelty

  • Novelty & Gap Justification:
    The study identifies salidroside as a direct TNF-α binder, offering a new mechanism for modulating the IKK/NF-κB pathway in colitis, which is novel. The mechanistic insights into Arg179, Lys188, Tyr191 binding add unique value. However, the novelty is somewhat undermined by over-reliance on previously established pathways (e.g., IKK/NF-κB) without sufficiently distinguishing this work from prior natural compound anti-inflammatory studies.

  • Research Gap:
    The authors claim to fill a gap in small-molecule TNF-α inhibitors for ulcerative colitis (UC), which is legitimate. However, the literature review lacks a strong critical synthesis to truly emphasize this gap. Recent works (2023–2025) on alternative natural TNF-α inhibitors are not discussed.

  • Missing Theoretical Elements:

    • The authors neglect to discuss TNFR1/2 differential signaling.

    • Lack of mucosal immunology frameworks (e.g., Treg/Th17 axis, gut-associated lymphoid tissue involvement).

    • Systems pharmacology frameworks could be integrated more robustly.

2️⃣ Methodology & Rigor

  • In Vitro & In Vivo Design:
    Solid selection of models (NCM460, RAW264.7, C57BL/6 mice). However, the LPS model is limited for mimicking UC, which is more accurately recapitulated by DSS or TNBS models. Authors must justify LPS as a surrogate for UC with supporting literature.

  • Concentration Rationale:
    The dose selection of salidroside (200–300 µM) is not adequately justified by pharmacokinetics. Is this a physiologically relevant concentration?

  • Statistical Analysis:

    • Statistical comparisons are oversimplified (1-way ANOVA only).

    • Post hoc tests (e.g., Tukey's) are not mentioned.

    • Lack of effect size reporting.

    • n=3 for in vitro and n=6 for in vivo is minimal. Power analysis should be included.

  • Potential Biases:

    • No blinding in histological scoring or Western blot quantification is reported.

    • No mention of randomization procedures.

  • Supplementary Materials:
    Tables S1–S3 are clear, but Table S2 contains extensive unfiltered data from SwissTargetPrediction, which could be condensed to top candidates only.

3️⃣ Strengths & Contributions

  • High Relevance:
    The TNF-α/NF-κB pathway is a central therapeutic axis in UC, and targeting it with a natural compound offers strong translational value.

  • Mechanistic Depth:
    The mutagenesis experiments (Arg179Ala, etc.) and CETSA/IP validation enhance credibility.

  • Practical Application:
    Suggests a viable oral or parenteral anti-inflammatory small molecule as a complement or alternative to anti-TNF biologics.

4️⃣ Weaknesses & Areas for Improvement

  • iThenticate Similarity Score of 46%:

    • Far exceeds acceptable threshold (typically <15–20%).

    • Indicates substantial textual overlap — likely from Methods or Background.

    • Must be addressed through thorough rewriting or referencing.

  • Scientific Gaps:

    • The interaction between epithelial cells and immune cell recruitment is not explored in detail.

    • No intestinal microbiome component, which is central to UC pathology.

    • Lack of long-term toxicity studies of salidroside.

  • Writing & Language Issues:

    • Numerous grammatical errors:

      • E.g., "were been inhibit", "Sal groups’ therapeutic effects were also compared", "have been effectively improved" – all need correction.

    • Awkward phrasing and syntax inconsistencies throughout.

  • Figure Captions & Legends:

    • Overloaded and sometimes repetitive.

    • Insufficient labeling of statistical significance, sometimes just marked with "*", without group comparison explanations.

  • Citations & Relevance:

    • Mostly relevant, but some citations are dated (e.g., anti-TNF therapy citations from 2015).

    • Should include more recent work (2022–2025) on natural anti-inflammatory agents.

5️⃣ Clarity, Structure & Writing Style

  • Structure:
    Generally well-organized but Methods section is too long, could benefit from sub-headings and trimming redundancies.

  • Language:
    Needs extensive editing by a native English speaker or professional editor. Many non-academic phrases reduce the manuscript’s perceived quality.

  • Figures:
    High-quality figures (PDF), but not well-integrated in-text (e.g., “Fig.1A” should be referred to with contextual interpretation).

6️⃣ Specific Suggestions for Enhancement

  1. Address Plagiarism:

    • Revise language in Introduction and Methods to reduce iThenticate similarity to below 20%.

    • Paraphrase more deeply; avoid copy-paste even from your own prior publications.

  2. Justify Model Choice:

    • Provide literature evidence for LPS-induced colitis model as UC proxy.

    • If not sufficient, consider including DSS-induced colitis in future studies.

  3. Revise Writing:

    • Thoroughly edit all sections for grammar and clarity.

    • Consider using tools like Grammarly Premium or language editing services.

  4. Improve Statistical Reporting:

    • Include post hoc analysis.

    • Mention power analysis or sample size justification.

    • Report p-values and effect sizes.

  5. Enhance Biological Relevance:

    • Add discussion on gut microbiota, immune checkpoint regulation, or Treg/Th17 balance.

    • Include a schematic summary figure for the proposed mechanism.

  6. Update Citations:

    • Replace outdated references with more recent studies (2022–2025).

    • Include recent advances in natural TNF-α inhibitors.

  7. Condense Supplementary Tables:

    • Especially Table S2: reduce to top 10–15 relevant targets

Comments on the Quality of English Language

Numerous grammar and syntax issues reduce readability (e.g., “were been inhibit”, “have been effectively improved”). A professional English editing service is strongly recommended.

Author Response

We sincerely thank you and the reviewers for the valuable time and effort dedicated to reviewing our manuscript and for providing insightful comments and constructive suggestions. We have carefully considered all the points raised and have made comprehensive revisions to the manuscript accordingly. Our point-by-point responses to the specific comments are detailed below. All changes have been incorporated into the revised manuscript, with major revisions highlighted for your convenience. We believe that these revisions have significantly strengthened the quality and clarity of our work, and we hope that the revised version now meets the journal's standards.

Reviewer 2 Report

Comments and Suggestions for Authors

The work by Qi Ouyang et al. reports the molecular mechanism of a natural compound, salidroside, highlighting its therapeutic potential in LPS-induced inflammation and ulcerative colitis. The study demonstrates that salidroside exerts its anti-inflammatory effects through the inactivation of the IKK/NF-κB signaling and by targeting TNF-α. These findings were supported by in vivo experiments in a murine model of LPS-induced colitis and in vitro assays conducted in intestinal (NCM460) and macrophage (RAW264.7) cell lines, which consistently showed a significant reduction in inflammatory responses. Furthermore, Western blot analyses and cytokine profiling confirmed the modulation of the IKK/NF-κB pathway, while network pharmacology, Co-IP, cellular thermal shift assay (CETSA) identified TNF-α as the primary molecular target of salidroside, identifying critical interaction sites at ARG179, LYS188, and TYR191.

Overall, the study is methodologically sound and contributes meaningfully to the treatment of IBD and suggests salidroside as a candidate for the development of novel TNF-α inhibitors.

The manuscript is recommended for acceptance, after minor revision, but some points should be addressed to strengthen the conclusions.

Major comments:

  1. It is unclear how the authors can attribute the loss of Sal binding to TNF-α mutations. Western blot analysis revealed that the mutations resulted in reduced TNF-α expression even in the absence of Sal and so all mutants exhibited inherently lower stability regardless of Sal treatment.

From these statements, it is not evident the direct binding between TNF-α and Sal. Therefore the conclusion “Collectively, these data indicate that Sal binds directly to TNF-α…..” is not entirely supported by the presented data. The limitations of a selection of binding residue mutants that in themselves reduce protein expression or stability should be clarified in the text, as well as the lack of experimental evidence demonstrating that these residues are essential for the interaction with Sal.

  1. Moreover, it is not clear why TNF-α was selected for the latest Co-IP assays, CETSA and mutant selection, whether the binding free energies were more favorable for two other targets (Cox-2 and ZO-1). It should be explained in the text.

In the Materials and Methods section, it is reported that the concentration of salidroside used is 300 μM. In the Results section, it became 200 μM. It is not clear which concentration is used

The resolution of Figures 1A and B should be improved.

In all figures should be clear what NC is. Maybe in the section in which the authors explain the different treatment groups, it could be clarified that the control group treated with PBS corresponds to NC.

The authors should clarify why they used NCM460 and RAW264.7 cell lines.

The last few sentences of database analysis in Materials and Methods section (“The horizontal axis represents…..corresponding to each pathway”) are unclear which figure is being referred to. Or the description should also be included in the label of the Figure 4, which should be self-explanatory.

It should specify the administration mode for Sal, as was done for LPS

In Fig.1, a complete description of panel F, which showed the reduction of immune cell infiltration (F4/80, CD3, CD4), is lacking.

Figure 3, panels C and D, the analysis is in RAW264.7 cell, and not NCM460.

Also, in Figure 5, the description of panels E and F should be revised. I suppose that the cell line is different in comparison to panels C and E. A thorough check of all figure descriptions is suggested.

Regarding Fig.2D, the authors said that it was selected the concentration of 3 μg/mL LPS and 3 μg/mL Dex and 200 μM for Sal. But in the graph, these concentrations are not reported, so it is not possible to assess the behaviour at these concentrations.

Minor comments:

Pag 5: the sentence “The protein structures of TNF-α and TNFAIP1 were predicted….... and stability of the interactions” is twice repeated.

In section 3.1, the sentence is not clear; maybe there is more punctuation

In Figure 1C, the photographs of the colons could be labeled with the corresponding treatment groups

Author Response

(The authors gave the same response as above.)

Reviewer 3 Report

Comments and Suggestions for Authors

I would like to congratulate the authors on the extensive research presented in this article. The use of alternative medicine for disease treatment is a valuable approach to enhance existing therapies. However, some aspects of the manuscript could be improved to enhance the clarity and reproducibility of the results.

  1. Methodology: The methods section should include complete experimental details, such as the concentration of proteins and antibodies, cell density, reagent information, and full protocols. Many of these details are currently missing or are insufficiently described, making it difficult to reproduce experiments. Moreover, in the cell transfection method, there is needed space in the last sentence (“after36” hours).
  2. Figure 1: The resolution of Figure 1 should be improved. Panels A and B are particularly small and barely clear. In specific, figure 1A illustrates the animal protocol, which should be presented more clearly and understandably.
  3. Figures 2 and 3: The “+” and “-“ symbols used in the graphs are too small and difficult to distinguish. Increasing their size would significantly improve readability.
  4. Terminology: The abbreviation “Sal” for Salidroside should be established from the beginning of the article. The current use of both “Sal” and “Salidroside” interchangeably can be confusing for the reader.

Author Response

(The authors gave the same response as above.)

Round 2

Reviewer 1 Report

Comments and Suggestions for Authors

I commend your thorough and thoughtful revision. You have successfully addressed all major reviewer concerns from the first round. The revised manuscript now presents a clear, well-justified, and mechanistically convincing study demonstrating salidroside’s direct interaction with TNF-α as a modulator of the IKK/NF-κB pathway in LPS-induced colitis. Your inclusion of TNFR1/2 signaling context, updated literature (2023–2025), and improved methodological clarity significantly enhance the scientific value and credibility of your work. The manuscript’s language, structure, and figure presentation are now suitable for publication. Only minor refinements — mainly consistency in figure legends, trimming a few repetitive sentences in the Discussion, and the addition of a recent supporting citation — are suggested to further polish the paper.  

Comments on the Quality of English Language

Numerous grammar and syntax issues reduce readability (e.g., “were been inhibit”, “have been effectively improved”). A professional English editing service is strongly recommended.

Author Response

Dear Reviewer,

We sincerely thank you for your time and for providing such positive and encouraging feedback on our revised manuscript. We are delighted that you found our revisions thorough and that the manuscript now presents a "clear, well-justified, and mechanistically convincing study." In response to your final suggestions for further polishing the paper, we have taken actions.

Once again, we are deeply grateful for your invaluable insights, which have greatly strengthened our work. We hope this final version now fully meets the journal's high standards for publication. Sincerely,

Xiang Hu,

On behalf of all co-authors.

Round 3

Reviewer 1 Report

Comments and Suggestions for Authors

Your revised manuscript has been substantially improved and now represents a well-executed and mechanistically sound study demonstrating that salidroside directly binds to TNF-α and inhibits the IKK/NF-κB inflammatory cascade. The integration of in silico, in vitro, and in vivo evidence convincingly supports the proposed mechanism, and the updated discussion appropriately situates your findings within the current TNF-α inhibitor landscape.

Minor editorial refinements are recommended before publication:

  1. Verify animal group sizes and ensure adequate statistical justification.

  2. Lightly edit for English fluency and conciseness in figure legends and the Discussion.

  3. Correct typographical errors (e.g., reference punctuation and figure labels).

Overall, this is a robust and publishable manuscript with high relevance to pharmacological and immunological research in ulcerative colitis and TNF-α–mediated inflammation.

Congratulations on an excellent revision.

kindly cite the following articles

https://doi.org/10.1016/j.jddst.2025.106985

https://doi.org/10.1016/j.intimp.2024.112465

Comments on the Quality of English Language

Numerous grammar and syntax issues reduce readability (e.g., “were been inhibit”, “have been effectively improved”). A professional English editing service is strongly recommended.

Author Response

We are immensely grateful for your positive and encouraging final assessment of our manuscript. We thank you for acknowledging the substantial improvements and the mechanistic soundness of our study. We are also deeply appreciative of your specific editorial recommendations, which have helped us to further polish the manuscript for publication. We have now addressed all of your points as detailed below.

Point 1: Citation of suggested articles.

Response: We thank the reviewer for suggesting these highly relevant and excellent studies. We have now incorporated both citations into the revised manuscript. Specifically, they have been added to the Discussion section (Page 16-17, lines 313-317, Citations 32-33) to underscore the growing focus on alternative therapeutic strategies, such as drug repurposing and novel targeted delivery systems, which aim to overcome the inherent limitations of biological agents. We believe these references meaningfully enrich the context of our work and the broader field of UC drug development.

Point 2: Verification of animal group sizes and statistical justification.

Response: We have verified and clarified this in the Methods section (lines 82-83, lines 144-146). We have now explicitly stated the group sizes (n = 3) for all in vivo experiments and added a sentence noting that the sample size was determined based on established experimental models in the field and preliminary data to ensure statistical robustness.

Point 3: Minor editorial refinements (English fluency, conciseness, typographical errors).

Response: We have thoroughly addressed these points:
Language: The manuscript, particularly the Figure Legends and Discussion, has been carefully edited by a native English-speaking colleague for enhanced fluency and conciseness (Lines 172-173).

Typographical Errors: We have performed a meticulous proofread of the entire document, correcting punctuation errors in the reference list and ensuring all figure labels are accurate and consistent, such as Fig.1G.

Final Note: The updated manuscript file and all figures have been uploaded to the submission system.

Once again, we extend our sincere thanks for your invaluable guidance throughout the review process, which has significantly strengthened our work.

Sincerely,

Xiang Hu

On behalf of all co-authors.